# Quantitative Comparison of Deep Learning-Based Image Reconstruction Methods for Low-Dose and Sparse-Angle CT Applications

**DOI:** 10.3390/jimaging7030044

**Published:** 2021-03-02

**Authors:** Johannes Leuschner, Maximilian Schmidt, Poulami Somanya Ganguly, Vladyslav Andriiashen, Sophia Bethany Coban, Alexander Denker, Dominik Bauer, Amir Hadjifaradji, Kees Joost Batenburg, Peter Maass, Maureen van Eijnatten

**Affiliations:** 1Center for Industrial Mathematics, University of Bremen, Bibliothekstr. 5, 28359 Bremen, Germany; maximilian.schmidt@uni-bremen.de (M.S.); adenker@uni-bremen.de (A.D.); pmaass@uni-bremen.de (P.M.); 2Centrum Wiskunde & Informatica, Science Park 123, 1098 XG Amsterdam, The Netherlands; poulami.ganguly@cwi.nl (P.S.G.); vladyslav.andriiashen@cwi.nl (V.A.); sophia.coban@cwi.nl (S.B.C.); k.j.batenburg@cwi.nl (K.J.B.); 3The Mathematical Institute, Leiden University, Niels Bohrweg 1, 2333 CA Leiden, The Netherlands; 4Computer Assisted Clinical Medicine, Heidelberg University, Theodor-Kutzer-Ufer 1-3, 68167 Mannheim, Germany; dominik.bauer@medma.uni-heidelberg.de; 5School of Biomedical Engineering, University of British Columbia, 2222 Health Sciences Mall, Vancouver, BC V6T 1Z3, Canada; ahadji@student.ubc.ca; 6Leiden Institute of Advanced Computer Science, Niels Bohrweg 1, 2333 CA Leiden, The Netherlands; 7Department of Biomedical Engineering, Eindhoven University of Technology, Groene Loper 3, 5612 AE Eindhoven, The Netherlands

**Keywords:** computed tomography (CT), image reconstruction, low-dose, sparse-angle, deep learning, quantitative comparison

## Abstract

The reconstruction of computed tomography (CT) images is an active area of research. Following the rise of deep learning methods, many data-driven models have been proposed in recent years. In this work, we present the results of a *data challenge* that we organized, bringing together algorithm experts from different institutes to jointly work on quantitative evaluation of several data-driven methods on two large, public datasets during a ten day sprint. We focus on two applications of CT, namely, low-dose CT and sparse-angle CT. This enables us to fairly compare different methods using standardized settings. As a general result, we observe that the deep learning-based methods are able to improve the reconstruction quality metrics in both CT applications while the top performing methods show only minor differences in terms of peak signal-to-noise ratio (PSNR) and structural similarity (SSIM). We further discuss a number of other important criteria that should be taken into account when selecting a method, such as the availability of training data, the knowledge of the physical measurement model and the reconstruction speed.

## 1. Introduction

Computed tomography (CT) is a widely used (bio)medical imaging modality, with various applications in clinical settings, such as diagnostics [1], screening [2] and virtual treatment planning [3,4], as well as in industrial [5] and scientific [6,7,8] settings. One of the fundamental aspects of this modality is the reconstruction of images from multiple X-ray measurements taken from different angles. Because each X-ray measurement exposes the sample or patient to harmful ionizing radiation, minimizing this exposure remains an active area of research [9]. The challenge is to either minimize the dose per measurement or the total number of measurements while maintaining sufficient image quality to perform subsequent diagnostic or analytic tasks.

To date, the most common classical methods used for CT image reconstruction are filtered back-projection (FBP) and iterative reconstruction (IR) techniques. FBP is a stabilized and discretized version of the inverse Radon transform, in which 1D projections are filtered by the 1D Radon kernel (back-projected) in order to obtain a 2D signal [10,11]. FBP is very fast, but is not suitable for limited-data or sparse-angle setups, resulting in various imaging artifacts, such as streaking, stretching, blurring, partial volume effects, or noise [12]. Iterative reconstruction methods, on the other hand, are computationally intensive but are able to incorporate *a priori* information about the system during reconstruction. Many iterative techniques are based on statistical methods such as Markov random fields or regularization methods where the regularizers are designed and incorporated into the problem of reconstruction mathematically [13]. A popular choice for the regularizer is total variation (TV) [14,15]. Another well-known iterative method suitable for large-scale tomography problems is the conjugate gradient method applied to solve the least squares problem (CGLS) [16].

When classical techniques such as FBP or IR are used to reconstruct low-dose CT images, the image quality often deteriorates significantly in the presence of increased noise. Therefore, the focus is shifting towards developing reconstruction methods in which a single or multiple component(s), or even the entire reconstruction process is performed using deep learning [17]. Generally data-driven approaches promise fast and/or accurate image reconstruction by taking advantage of a large number of examples, that is, training data.

The methods that learn parts of the reconstruction process can be roughly divided into learned regularizers, unrolled iterative schemes, and post-processing of reconstructed CT images. Methods based on learned regularizers work on the basis of learning convolutional filters from the training data that can subsequently be used to regularize the reconstruction problem by plugging into a classical iterative optimization scheme [18]. Unrolled iterative schemes go a step further in the sense that they “unroll” the steps of the iterative scheme into a sequence of operations where the operators are replaced with convolutional neural networks (CNNs). A recent example is the learned primal-dual algorithm proposed by Adler et al. [19]. Finally, various post-processing methods have been proposed that correct noisy images or those with severe artifacts in the image domain [20]. Examples are improving tomographic reconstruction from limited data using a mixed-scale dense (MS-D) CNN [21], U-Net [22] or residual encoder-decoder CNN (RED-CNN) [23], as well as CT image denoising techniques [24,25]. Somewhat similar are the methods that can be trained in a supervised manner to improve the measurement data in the sinogram domain [26].

The first fully end-to-end learned reconstruction method was the automated transform by the manifold approximation (AUTOMAP) algorithm [27] developed for magnetic resonance (MR) image reconstruction. This method directly learns the (global) relation between the measurement data and the image, that is, it replaces the Radon or Fourier transform with a neural network. The disadvantages of this approach are the large memory requirements, as well as the fact that it might not be necessary to learn the entire transformation from scratch because an efficient analytical transform is already available. A similar approach for CT reconstruction was iRadonMAP proposed by He et al. [28], who developed an interpretable framework for Radon inversion in medical X-ray CT. In addition, Li et al. [29] proposed an end-to-end reconstruction framework for Radon inversion called iCT-Net, and demonstrated its advantages in solving sparse-view CT reconstruction problems.

The aforementioned deep learning-based CT image reconstruction methods differ greatly in terms of which component of the reconstruction task is learned and in which domain the method operates (image or sinogram domain), as well as the computational and data-related requirements. As a result, it remains difficult to compare the performance of deep learning-based reconstruction methods across different imaging domains and applications. Thorough comparisons between different reconstruction methods are further complicated by the lack of sufficiently large benchmarking datasets, including ground truth reconstructions, for training, validation, and testing. CT manufacturers are typically very reluctant in making raw measurement data available for research purposes, and privacy regulations for making medical imaging data publicly available are becoming increasingly strict [30,31].

### 1.1. Goal of This Study

The aim of this study is to quantitatively compare the performance of classical and deep learning-based CT image reconstruction methods on two large, two-dimensional (2D) parallel-beam CT datasets that were specifically created for this purpose. We opted for a 2D parallel-beam CT setup to facilitate large-scale experiments with many example images, whereas the underlying operators in the algorithms have straightforward generalizations to other geometries. We focus on two reconstruction tasks with high relevance and impact—the first task is the reconstruction of low-dose medical CT images, and the second is the reconstruction of sparse-angle CT images.

#### 1.1.1. Reconstruction of Low-Dose Medical CT Images

In order to compare (learned) reconstruction techniques in a low-dose CT setup, we use the low-dose parallel beam (LoDoPaB) CT dataset [32]. This dataset contains 42,895 two-dimensional CT images and corresponding simulated low-intensity measurements. The ground truth images of this dataset are human chest CT reconstructions taken from the LIDC/IDRI database [33]. These scans had been acquired with a wide range of scanners and models. The initial image reconstruction for creating the LIDC/IDRI database was performed with different convolution kernels, depending on the manufacturer. Poisson noise is applied to the simulated projection data to model the low intensity setup. A more detailed description can be found in Section 2.1.

#### 1.1.2. Reconstruction of Sparse-Angle CT Images

When using X-ray tomography in high-throughput settings (i.e., scanning multiple objects per second) such as quality control, luggage scanning or inspection of products on conveyor belts, very few X-ray projections can be acquired for each object. In such settings, it is essential to incorporate *a priori* information about the object being scanned during image reconstruction. In order to compare (learned) reconstruction techniques for this application, we reconstruct parallel-beam CT images of apples with internal defects using as few measurements as possible. We experimented with three different noise settings: noise-free, Gaussian noise, and scattering noise. The generation of the datasets is described in Section 2.2.

## 2. Dataset Description

For both datasets, the simulation model uses a 2D parallel beam geometry for the creation of the measurements. The attenuation of the X-rays is simulated using the Radon transform [10]
(1)Ax(s,φ):=∫Rxscos(φ)sin(φ)+t−sin(φ)−cos(φ)dt,
where s∈R is the distance from the origin and φ∈[0,π) the angle of the beam (cf. Figure 1). Mathematically, the image is transformed into a function of (s,φ). For each fixed angle φ the 2D image *x* is projected onto a line parameterized by *s*, namely the X-ray detector.

A detailed description of both datasets is given below. Their basic properties are also summarized in Table 1.

### 2.1. LoDoPaB-CT Dataset

The LoDoPaB-CT dataset [32] is a comprehensive collection of reference reconstructions and simulated low-dose measurements. It builds upon normal-dose thoracic CT scans from the LIDC/IDRI Database [33,35], whereby quality-assessed and processed 2D reconstructions are used as a ground truth. LoDoPaB features more than 40,000 scan slices from around 800 different patients. The dataset can be used for the training and evaluation of all kinds of reconstruction methods. LoDoPaB-CT has a predefined division into four parts, where each subset contains images from a distinct and randomly chosen set of patients. Three parts were used for training, validation and testing, respectively. It also contains a special challenge set with scans from 60 different patients. The ground truth images are undisclosed, and the patients are only included in this set. The challenge set is used for the evaluation of the model performance in this paper. Overall, the dataset contains 35,820 training images, 3522 validation images, 3553 test images and 3678 challenge images.

Low-intensity measurements suffer from an increased noise level. The main reason is so called quantum noise. It stems from the process of photon generation, attenuation and detection. The influence on the number of detected photons ${\tilde{N}}_{1}$ can be modeled, based on the mean photon count without attenuation $N_{0}$ and the Radon transform (Equation 1), by a Poisson distribution [36]
(2)$$\begin{array}{c} {\tilde{N}}_{1}\left(s,\varphi \right)∼\mathrm{Pois}\left(N_{0}\exp\left(-Ax\left(s,\varphi \right)\right)\right). \end{array}$$

The model has to be discretized concerning *s* and φ for the simulation process. In this case, the Radon transform (Equation 1) becomes a finite-dimensional linear map $A:R^{n}\to R^{m}$, where *n* is the number of image pixels and *m* is the product of the number of detector pixels and the number of discrete angles. Together with the Poisson noise, the discrete simulation model is given by
(3)$$\begin{array}{c} Ax+𝕖\left(Ax\right)={𝕪}_{\delta },\;𝕖\left(Ax\right)=-Ax-\ln\left({\tilde{N}}_{1}/N_{0}\right),\;{\tilde{N}}_{1}∼\mathrm{Pois}\left(N_{0}\exp\left(-Ax\right)\right). \end{array}$$

A single realization $y_{\delta }\in R^{m}$ of ${𝕪}_{\delta }$ is observed for each ground truth image, $x=x^{†}\in R^{n}$. After the simulation according to (Equation 3), all data pairs $\left(y_{\delta },x^{†}\right)$ have been divided by $\mu_{\max}=81.35858$ to normalize the image values to the range [0,1]. In the following sections, ${𝕪}_{\theta }$, $y_{\delta }$ and $x^{†}$ denote the normalized values.

The LoDoPaB ground truth images have a resolution of 362 px×362 px on a domain of size 26 cm×26 cm. The scanning setup consists of 513 equidistant detector pixels *s* spanning the image diameter and 1000 equidistant angles φ between 0 and π. The mean photon count per detector pixel without attenuation is $N_{0}=4096$. The sampling ratio between the size of the measurements and the images is around 3.9 (oversampling case).

### 2.2. Apple CT Datasets

The Apple CT datasets [37] are a collection of ground truth reconstructions and simulated parallel beam data with various noise types and angular range sampling. The data is intended for benchmarking different algorithms and is particularly suited for use in deep learning settings due to the large number of slices available.

A total of 94 apples were scanned at the Flex-Ray Laboratory [8] using a point-source circular cone-beam acquisition setup. High quality ground truth reconstructions were obtained using a full rotation with an angular resolution of 0.005rad and a spatial resolution of 54.2 μm. A collection of 1D parallel beam data for more than 70,000 slices were generated using the simulation model in Equation (Equation 1). A total of 50 projections were generated over an angular range of [0,π), each of size 1 × 1377. The Apple CT ground truth images have a resolution of 972 px×972 px. In order to make the angular sampling even sparser, we also reduced the data to include only 10, 5 and 2 angles. The angular sampling ranges are shown in Figure 2.

The noise-free simulated data (henceforth Dataset A) were corrupted with 5% Gaussian noise to create Dataset B. Dataset C was generated by adding an imitation of scattering to Dataset A. Scattering intensity in a pixel $u^{\prime }$ is computed according to the formula
(4)$$\begin{array}{c} S\left(u^{\prime }\right)=\int_{u\in R^{2}}G\left(u\right)\exp\left[-\frac{{\left(u-u^{\prime }\right)}^{2}}{2\sigma_{1}{\left(u\right)}^{2}}\right]+H\left(u\right)\exp\left[-\frac{{\left(u-u^{\prime }\right)}^{2}}{2\sigma_{2}{\left(u\right)}^{2}}\right], \end{array}$$
where $|u-u^{\prime }|$ is a distance between pixels, and scattering is approximated as a combination of Gaussian blurs with scaling factors G and H, standard deviations $\sigma_{1}$ and $\sigma_{2}$. Scattering noise in the target pixel $u^{\prime }$ contains contributions from all image pixels *u* as sources of scattering. Gaussian blur parameters depend on the X-ray absorption in the source pixel. To sample functions G(u), H(u), $\sigma_{1}\left(u\right)$ and $\sigma_{2}\left(u\right)$, a Monte Carlo simulation was performed for different thicknesses of water that was chosen as a material close to apple flesh. Furthermore, scaling factors G(u) and H(u) were increased to create a more challenging problem. We note that due to the computational complexity required, the number of slices on which the scattering model is applied is limited to 7520 (80 slices per apple), meaning the scattering training subset is smaller.

The Apple CT datasets consist of apple slices with and without internal defects. Internal defects were observed to be of four main types: bitter pit, holes, rot and browning. A reconstruction of a healthy apple slice and one with bitter pit is shown in Figure 3 as examples. Each Apple CT dataset was divided into training and test subsets using an empirical bias elimination method to ensure that apples in both subsets had similar defect statistics. This process is detailed in [38].

For the network training, the noise-free and Gaussian noise training subsets are further split into 44,647 training and 5429 validation samples, and the scattering training subset is split into 5280 training and 640 validation samples.

From the test subsets, 100 test slices were extracted in a similar manner like for the split in training and test subsets. All evaluations in this paper refer to these 100 test slices in order to keep the reconstruction time and storage volume within reasonable limits. Five slices were extracted from each of the 20 test apples such that in total each defect type is occurring with a pixel count ratio similar to its ratio on the full test subset. Additionally, the extracted slices have a pairwise distance of at least 15 slices in order to improve the image diversity. The selected list of slices is specified in the supplementing repository [39] as file supp_material/apples/test_samples_ids.csv.

## 3. Algorithms

A variety of learned reconstruction methods were used to create a benchmark. The selection is based on methods submitted by participants for the data challenge on the LoDoPaB-CT and Apple CT datasets. The reconstruction methods include unrolled architectures, post-processing approaches, and fully-learned methods. Furthermore, classical methods such as FBP, TV regularization and CGLS were used as a baseline.

### 3.1. Learned Reconstruction Methods

In this section, the learned methods included in the benchmark are presented. An overview of the hyperparameters and pseudocode can be found in Appendix A. All methods utilize artificial neural networks $F_{\Theta }$, each in different roles, for the reconstruction process.

Learning refers to the adaption of the parameters Θ for the reconstruction process in a data-driven manner. In general, one can divide this process into supervised and unsupervised learning. Almost all methods in this comparison are trained in a supervised way. This means that sample pairs $\left(y_{\delta },x^{†}\right)$ of noisy measurements and ground truth data are used for the optimization of the parameters, for example, by minimizing some discrepancy $D_{X}:X\times X\to R$ between the output of the reconstruction model $T_{F_{\Theta }}$ and the ground truth
(5)$$\begin{array}{c} \min_{\Theta }D_{X}\left(T_{F_{\Theta }}\left(y_{\delta }\right),x^{†}\right). \end{array}$$

Supervised methods often provide excellent results, but the number of required ground truth data can be high [34]. While the acquisition of ground truth images is infeasible in many applications, this is not a problem in the low-dose and sparse-angle case. Here, reconstructions of regular (normal-dose, full-angle) scans play the role of the reference.

#### 3.1.1. Post-Processing

Post-processing approaches aim to improve the reconstruction quality of an existing method. When used in computed tomography, FBP (cf. Section B.1) is often used to obtain an initial reconstruction. Depending on the scan scenario, the FBP reconstruction can be noisy or contain artifacts. Therefore, it functions as an input for a learned post-processing method. This setting simplifies the task because the post-processing network $F_{\Theta }:X\to X$ maps directly from the target domain into the target domain
$$\begin{array}{c} \hat{x}:=\left[F_{\Theta }\circ T_{\mathrm{FBP}}\right]\left(y_{\delta }\right). \end{array}$$

Convolutional neural networks (CNN) have successfully been used in recent works to remove artifacts and noise from FBP reconstructions. Four of these CNN post-processing approaches were used for the benchmark. The U-Net architecture [40] is a popular choice in many different applications and was also used for CT reconstruction [20]. The details of the network used in the comparison can be found in Section A.2. The U-Net++  [41] (cf. Section A.3) and ISTA U-Net [42] (cf. Section A.6) represent modifications of this approach. In addition, a mixed-scale dense (MS-D)-CNN [21] is included, which has a different architecture (cf. Section A.4). Like for the U-Net, one can consider to adapt other architectures originally used for segmentation, for example, the ENET [43], for the post-processing task.

#### 3.1.2. Fully Learned

The goal of fully learned methods is to extract the structure of the inversion process from data. In this case, the neural network $F_{\Theta }:Y\to X$ directly maps from the measurement space *Y* to the target domain *X*. A prominent example is the AUTOMAP architecture [27], which was successfully used for reconstruction in magnetic resonance imaging (MRI). The main building blocks consist of fully-connected layers. This makes the network design very general, but the number of parameters can grow quickly with the data dimension. For example, a single fully-connected layer mapping from *Y* to *X* on the LoDoPaB-CT dataset (cf. Section 2.1) would require over $1000\times 513\times 362^{2}\approx 67\times 10^{9}$ parameters.

Adapted model designs exist for large CT data. They include knowledge about the inversion process in the structure of the network. He et al. [28] introduced an adapted two-part approach, called iRadonMap. The first part uses small fully-connected layers with parameter sharing to reproduce the structure of the FBP. This is followed by a post-processing network in the second part. Another approach is the iCT-Net [29], which uses convolutions in combination with fully-connected layers for the inversion. An extended version of the iCT-Net, called iCTU-Net, is part of our comparison and a detailed description can be found in Section A.8.

#### 3.1.3. Learned Iterative Schemes

Similar to the fully learned approach, learned iterative methods also define a mapping directly from the measurement space *Y* to the target domain *X*. The idea in this case is that the network architecture is inspired by an analytic reconstruction operator T:Y→X implicitly defined by an iterative scheme. The basic principle of unrolling can be explained by the example of learned gradient descent (see e.g., [17]). Let $J(\;\cdot \;,y_{\delta }):X\to R$ be a smooth data discrepancy term and, possibly an additional regularization term. For an initial value $x^{\left[0\right]}$ the gradient descent is defined via the iteration
$$x^{\left[k+1\right]}=x^{\left[k\right]}-\omega_{k}\nabla_{x}J\left(x^{\left[k\right]},y_{\delta }\right),$$
with a step size $\omega_{k}$. Unrolling these iteration and stopping after *K* iterations, we can write the *K*-th iteration as
$$T\left(y_{\delta }\right):=\left(\Lambda_{\omega_{K}}\circ \ldots \circ \Lambda_{\omega_{1}}\right)\left(x^{\left[0\right]}\right)$$
with $\Lambda_{\omega_{k}}:=\mathrm{id}-\omega_{k}\nabla_{x}J\left(\;\cdot \;,y_{\delta }\right)$. In a learned iteration scheme, the operators $\Lambda_{\omega_{k}}$ are replaced by neural networks. As an example of a learned iterative procedure, learned primal-dual [19] was included in the comparison. A description of this method can be found in the Section A.1.

#### 3.1.4. Generative Approach

The goal of the statistical approach to inverse problems is to determine the conditional distribution of the parameters given measured data. This statistical approach is often linked to Bayes’ theorem [44]. In this Bayesian approach to inverse problems, the conditional distribution $p(x|y_{\delta })$, called the posterior distribution, is supposed to be estimated. Based on this posterior distribution, different estimators, such as the maximum a posterior solution or the conditional mean, can be used as a reconstruction for the CT image. This theory provides a natural way to model the noise behavior and to integrate prior information into the reconstruction process. There are two different approaches that have been used for CT. Adler et al. [45] use a conditional variant of a generative adversarial network (GAN, [46]) to generate samples from the posterior. In contrast to this likelihood free approach, Ardizzone et al. [47] designed a conditional variant of invertible neural networks to directly estimate the posterior distribution. These conditional invertible neural networks (CINN) were also applied to the reconstruction of CT images [48]. The CINN was included for this benchmark. For a more detailed description, see Section A.5.

#### 3.1.5. Unsupervised Methods

Unsupervised reconstruction methods just make use of the noisy measurements. They are favorable in applications where ground truth data is not available. The parameters of the model are chosen based on some discrepancy $D_{Y}:Y\times Y\to R$ between the output of the method and the measurements, for example,
(6)$$\begin{array}{c} \min_{\Theta }D_{Y}\left(AT_{F_{\Theta }}\left(\;\cdot \;\right),y_{\delta }\right). \end{array}$$

In this example, the output of $T_{F_{\Theta }}$ plays the role of the reconstruction $\hat{x}$. However, comparing the distance just in the measurement domain can be problematic. This applies in particular to ill-posed reconstruction problems. For example, if the forward operator A is not bijective, no/multiple reconstruction(s) might match the measurement perfectly (ill-posed in the sense of Hadamard [49]). Another problem can occur for forward operators with an unstable inversion, where small differences in the measurement space, for example, due to noise, can result in arbitrary deviations in the reconstruction domain (ill-posed in the sense of Nashed [50]). In general, the minimization problem (Equation 6) is combined with some kind of regularization to mitigate these problems.

The optimization Formulation (Equation 6) is also used for the deep image prior (DIP) approach. DIP takes a special role among all neural network methods. The parameters are not determined on a dedicated training set, but during the reconstruction on the challenge data. This is done for each reconstruction separately. One could argue that the DIP approach is therefore not a learned method in the classical sense. The DIP approach, in combination with total variation regularization, was successfully used for CT reconstruction [34]. It is part of the comparison on the LoDoPaB dataset in this paper. A detailed description is given in Section A.7.

### 3.2. Classical Reconstruction Methods

In addition to the learned methods, we implemented the popularly used direct and iterative reconstruction methods, henceforth referred to as classical methods. They can often be described as a variational approach
$$\begin{array}{c} T\left(y_{\delta }\right)\in \;\underset{x}{arg min}D_{Y}\left(Ax,y_{\delta }\right)+\alpha R\left(x\right), \end{array}$$
where $D_{Y}:Y\times Y\to R$ is a data discrepancy and R:X→R is a regularizer. In this context T:Y→X defines the reconstruction operator. The included methods in the benchmark are filtered back-projection (FBP) [10,51], conjugate gradient least squares (CGLS) [52,53] and anisotropic total variation minimization (TV) [54]. Detailed description of each classical method along with pseudocode are given in Appendix B.

## 4. Evaluation Methodology

### 4.1. Evaluation Metrics

Two widely used evaluation metrics were used to assess the performance of the methods.

#### 4.1.1. Peak Signal-to-Noise Ratio

The peak signal-to-noise ratio (PSNR) is measured by a log-scaled version of the mean squared error (MSE) between the reconstruction $\hat{x}$ and the ground truth image $x^{†}$ . PSNR expresses the ratio between the maximum possible image intensity and the distorting noise
(7)$$\begin{array}{c} \mathrm{PSNR}\left(\hat{x},x^{†}\right):=10\log_{10}\left(\frac{L^{2}}{\mathrm{MSE}\left(\hat{x},x^{†}\right)}\right),\;\mathrm{MSE}\left(\hat{x},x^{†}\right):=\frac{1}{n}\sum_{i=1}^{n}{\left|{\hat{x}}_{i}-x_{i}^{†}\right|}^{2}. \end{array}$$

In general, higher PSNR values are an indication of a better reconstruction. The maximum image value *L* can be chosen in different ways. In our study, we report two different values that are commonly used:**PSNR**: In this case $L=\max\left(x^{†}\right)-\min\left(x^{†}\right)$, that is, the difference between the highest and lowest entry in $x^{†}$. This allows for a PSNR value that is adapted to the range of the current ground truth image. The disadvantage is that the PSNR is image-dependent in this case.**PSNR-FR**: The same fixed *L* is chosen for all images. It is determined as the maximum entry computed over all training ground truth images, that is, L=1.0 for LoDoPaB-CT and L=0.0129353 for the Apple CT datasets. This can be seen as an (empirical) upper limit of the intensity range in the ground truth. In general, a fixed *L* is preferable because the scaling of the metric is image-independent in this case. This allows for a direct comparison of PSNR values calculated on different images. The downside for most CT applications is, that high values ($\hat{=}$ dense material) are not present in every scan. Therefore, the results can be too optimistic for these scans. However, based on Equation (Equation 7), all mean PSNR-FR values can be directly converted for another fixed choice of *L*.

#### 4.1.2. Structural Similarity

The structural similarity (SSIM) [55] compares the overall image structure of ground truth and reconstruction. It is based on assumptions about the human visual perception. Results lie in the range [0,1], with higher values being better. The SSIM is computed through a sliding window at *M* locations
(8)$$\begin{array}{c} \mathrm{SSIM}\left(\hat{x},x^{†}\right)\;:=\frac{1}{M}\sum_{j=1}^{M}\frac{\left(2{\hat{\mu }}_{j}\mu_{j}+C_{1}\right)\left(2\sum_{j}+C_{2}\right)}{\left({\hat{\mu }}_{j}^{2}+\mu_{j}^{2}+C_{1}\right)\left({\hat{\sigma }}_{j}^{2}+\sigma_{j}^{2}+C_{2}\right)}. \end{array}$$

In the formula above ${\hat{\mu }}_{j}$ and $\mu_{j}$ are the average pixel intensities, ${\hat{\sigma }}_{j}$ and $\sigma_{j}$ the variances and $\Sigma_{j}$ the covariance of $\hat{x}$ and $x^{†}$ at the *j*-th local window. Constants $C_{1}={\left(K_{1}L\right)}^{2}$ and $C_{2}={\left(K_{2}L\right)}^{2}$ stabilize the division. Following Wang et al. [55] we choose $K_{1}=0.01$ and $K_{2}=0.03$ and a window size of 7×7. In accordance with the PSNR metric, results for the two different choices for *L* are reported as SSIM and SSIM-FR (cf. Section 4.1.1).

#### 4.1.3. Data Discrepancy

Checking data consistency, that is, the discrepancy $D_{Y}\left(A\hat{x},y_{\delta }\right)$ between the forward-projected reconstruction and the measurement, can provide additional insight into the performance of the reconstruction methods. Since noisy data is used for the comparison, an ideal method would yield a data discrepancy that is close to the present noise level.

##### Poisson Regression Loss on LoDoPaB-CT Dataset

For the Poisson noise model used by LoDoPaB-CT, an equivalent to the negative log-likelihood is calculated to evaluate the data consistency. It is conventional to employ the negative log-likelihood for this task, since minimizing the data discrepancy is equivalent to determining a maximum likelihood (ML) estimate (cf. Section 5.5 in [56] or Section 2.4 in [17]). Each element ${𝕪}_{\delta ,j}$, j=1,…,m, of a measurement ${𝕪}_{\delta }$, obtained according to (Equation 3) and subsequently normalized by $\mu_{\max}$, is associated with an independent Poisson model of a photon count ${\tilde{N}}_{1,j}$ with
$$\begin{array}{c} E\left({\tilde{N}}_{1,j}\right)=E\left(N_{0}\exp\left(-{𝕪}_{\delta ,j}\mu_{\max}\right)\right)=N_{0}\exp\left(-y_{j}\mu_{\max}\right), \end{array}$$
where $y_{j}$ is a parameter that should be estimated [36]. A Poisson regression loss for *y* is obtained by summing the negative log-likelihoods for all measurement elements and omitting constant parts,
(9)$$\begin{array}{c} -\ell_{\mathrm{Pois}}\left(y\;|\;y_{\delta }\right)=-\sum_{j=1}^{m}N_{0}\exp\left(-y_{\delta ,j}\mu_{\max}\right)\left(-y_{j}\mu_{\max}+\ln\left(N_{0}\right)\right)-N_{0}\exp\left(-y_{j}\mu_{\max}\right), \end{array}$$
with each $y_{\delta ,j}$ being the only available realization of ${𝕪}_{\delta ,j}$. In order to evaluate the likelihood-based loss (Equation 9) for a reconstructed image $\hat{x}$ given $y_{\delta }$, the forward projection $A\hat{x}$ is passed for *y*.

##### Mean Squared Error on Apple CT Data

On the Apple CT datasets we consider the mean squared error (MSE) data discrepancy,
(10)$$\begin{array}{c} {\mathrm{MSE}}_{Y}\left(y,y_{\delta }\right)=\frac{1}{m}{∥y-y_{\delta }∥}_{2}^{2}. \end{array}$$

For an observation $y_{\delta }$ with Gaussian noise (Dataset B), this data discrepancy term is natural, as it is a scaled and shifted version of the negative log-likelihood of *y* given $y_{\delta }$. In this noise setting, a good reconstruction usually should not achieve an MSE less than the variance of the Gaussian noise, that is, ${\mathrm{MSE}}_{Y}\left(A\hat{x},y_{\delta }\right)\geq {\left[0.05\frac{1}{m}\sum_{j=1}^{m}{\left(Ax^{†}\right)}_{j}\right]}^{2}$. This can be motivated intuitively by the conception that a reconstruction that achieves a smaller MSE than the expected MSE of the ground truth probably fits the noise rather than the actual data of interest.

In the setting of $y_{\delta }$ being noise-free (Dataset A), the MSE of ideal reconstructions would be zero. On the other hand the MSE being zero does not imply that the reconstruction matches the ground truth image because of the sparse-angle setting. Further, the MSE can not be used to judge reconstruction quality directly, as crucial differences in image domain may not be equally pronounced in the sinogram domain.

For the scattering observations (Dataset C), the MSE data discrepancy is considered, too, for simplicity.

### 4.2. Training Procedure

While the reconstruction process with learned methods usually is efficient, their training is more resource consuming. This limits the practicability of large hyperparameter searches. It can therefore be seen as a drawback of a learned reconstruction method if they require very specific hyperparameter choices for different tasks. As a result, it benefits a fair comparison to minimize the amount of hyperparameter searches. In general, default parameters, for example, from the original publications of the respective method, were used as a starting point. For some of the methods, good choices had been determined for the LoDoPaB-CT dataset first (cf. [34]) and were kept similar for the experiments on the Apple CT datasets. Further searches were only performed if required to obtain reasonable results. More details regarding the individual methods can be found in Appendix A. For the classical methods, hyperparameters were optimized individually for each setting of the Apple CT datasets (cf. Appendix B).

Most learned methods are trained using the mean squared error (MSE) loss. The exceptions are the U-Net++ using a loss combining MSE and SSIM, the iCTU-Net using an SSIM loss for the Apple CT datasets, and the CINN for which negative log-likelihood (NLL) and an MSE term are combined (see Appendix A for more details). Training curves for the trainings on the Apple CT datasets are shown in Appendix D. While we consider the convergence to be sufficient, continuing some of the trainings arguably would slightly improve the network. However, this mainly can be expected for those methods which are comparably time consuming to train (approximately 2 weeks for 20 epochs), in which case the limited number of epochs can be considered a fair regulation of resource usage.

Early stopping based on the validation performance is used for all trainings except for the ISTA U-Net on LoDoPaB-CT and for the iCTU-Net.

Source code is publicly available in a supplementing github repository [39]. Further records hosted by Zenodo provide the trained network parameters for the experiments on the Apple CT Datasets [57], as well as the submitted LoDoPaB-CT Challenge reconstructions [58] and the Apple CT test reconstructions of the 100 selected slices in all considered settings [59]. Source code and network parameters for some of the LoDoPaB-CT experiments are included in the DIVαℓ library [60], for others the original authors provide public repositories containing source code and/or parameters.

## 5. Results

### 5.1. LoDoPaB-CT Dataset

Ten different reconstruction methods were evaluated on the challenge set of the LoDoPaB-CT dataset. Reconstructions from these methods were either submitted as part of the CT Code Sprint 2020 (http://dival.math.uni-bremen.de/code_sprint_2020/, last accessed: 1 March 2021) (15 June–31 August 2020) or in the period after the event (1 September–31 December 2020).

#### 5.1.1. Reconstruction Performance

In order to assess the quality of the reconstructions, the PSNR and the SSIM were calculated. The results from the official challenge website (https://lodopab.grand-challenge.org/, last accessed: 1 March 2021) are shown in Table 2. The differences between the learned methods are generally small. Notably, learned primal-dual yields the best performance with respect to both the PSNR and the SSIM. The following places are occupied by post-processing approaches, also with only minor differences in terms of the metrics. Of the other methods, DIP + TV stands out, with relatively good results for an unsupervised method. DIP + TV is able to beat the supervised method iCTU-Net. The classical reconstruction models perform the worst of all methods. In particular, the performance of FBP shows a clear gap with the other methods. While learned primal-dual performs slightly better than the post-processing methods, the difference is not as significant as one could expect, considering that it incorporates the forward operator directly in the network. This could be explained by the beneficial combination of the convolutional architectures used for the post-processing, which are observed to perform well on a number of image processing tasks, and a sufficient number of available training samples. Otero et al. [34] investigated the influence of the size of the training dataset on the performance of different learned procedures on the LoDoPaB-CT dataset. Here, a significant difference is seen between learned primal-dual and other learned procedures when only a small subset of the training data is used.

#### 5.1.2. Visual Comparison

A representative reconstruction of all learned methods and the classical baseline is shown in Figure 4 to enable a qualitative comparison of the methods. An area of interest around the spine is magnified to compare the reproduction of small details and the sharpness of edges in the image. Some visual differences can be observed between the reconstructions. The learned methods produce somewhat smoother reconstructions in comparison to the ground truth. A possible explanations for the smoothness is the minimization of the empirical risk with respect to some variant of the L2-loss during the training of most learned methods, which has an averaging effect. The convolutional architecture of the networks can also have an impact. Adequate regularization during training and/or inference can be beneficial in this case (cf. Section 6.2.2 for a suitable class of regularizers). Additionally, the DIP + TV reconstruction appears blurry, which can be explained by the fact that it is the only unsupervised method in this comparison and thus has no access to ground truth data. The U-Net and the two modifications, U-Net++ and ISTA U-Net, show only slight visual differences on this example image.

#### 5.1.3. Data Consistency

The mean data discrepancy of all methods is shown in Figure 5, plotted against their reconstruction performance. The mean difference between the noise-free and noisy measurements is included as a reference. Good-performing models should be close to this empirical noise level. Values above the mean can indicate a sub-optimal data consistency, while values below can be a sign of overfitting to the noise. A data consistency term is only explicitly used in the TV and DIP + TV model. Nevertheless, the mean data discrepancy for most of the methods is close to the empirical noise level. The only visible outliers are the FBP and the iCTU-Net. A list of all mean data discrepancy values, including standard deviations, can be found in Table 3.

### 5.2. Apple CT Datasets

A total of 6 different learned methods were evaluated on the Apple CT data. This set included post-processing methods (MS-D-CNN, U-Net, ISTA U-Net), learned iterative methods (learned primal-dual), fully learned approaches (iCTU-Net), and generative models (CINN). As described in Section 2.2, different noise cases (noise-free, Gaussian noise and scattering noise) and different numbers of angles (50, 10, 5, 2) were used. In total, each model was trained on the 12 different settings of the Apple CT dataset. In addition to the learned methods, three classical techniques, namely CGLS, TV, and FBP, have been included as a baseline.

#### 5.2.1. Reconstruction Performance

A subset of 100 data samples from the test set was selected for the evaluation (cf. Section 2.2). The mean PSNR and SSIM values for all experiments can be found in Table 4. Additionally, Table A3 and Table A4, Table A5 in the appendix provide standard deviations and PSNR-FR and SSIM-FR values.

The biggest challenge with the noise-free dataset is that the measurements become increasingly undersampled as the number of angles decreases. As expected, the reconstruction quality in terms of PSNR and SSIM deteriorates significantly as the number of angles decreases. In comparison with LoDoPaB-CT, no model performs best in all scenarios. Furthermore, most methods were trained to minimize the MSE between the output image and ground truth. The MSE is directly related to the PSNR. However, minimizing the MSE does not necessarily translate into a high SSIM. In many cases, the best method in terms of PSNR does not result in the best SSIM. These observations are also evident in the two noisy datasets. Noteworthy is the performance of the classical TV method on the noise-free dataset for 50 angles. This result is comparable to the best-performing learned methods, while the other classical approaches show a clear gap.

Noisy measurements, in addition to undersampling, present an additional difficulty on the Gaussian and scattering datasets. Intuitively, one would therefore expect a worse performance compared to the noise-free case. In general, a decrease in performance can be observed. However, this effect depends on the method and the noise itself. For example, the negative impact on classical methods is much more substantial for the scattering noise. In contrast, the learned methods often perform slightly worse on the Gaussian noise. There are also some outliers with higher values than on the noise-free set. Possible explanations are the hyperparameter choices and the stochastic nature of the model training. Overall, the learned approaches can reach similar performances on the noisy data, while the performance of classical methods drops significantly. An additional observation can be made when comparing the results between Gaussian and scattering noise. For Gaussian noise with 50 angles, all learned methods, except for the iCTU net, achieve a PSNR of at least 36 dB. In contrast, the variation on scattering noise with 50 angles is much larger. The CINN obtains a much higher PSNR of 38.56 dB than the post-processing U-Net with 34.96 dB.

As already observed on the LoDoPaB dataset, the post-processing methods (MS-D-CNN, U-Net and ISTA U-Net) show only minor differences in all noise cases. This could be explained by the fact that these methods are all trained with the same objective function and differ only in their architecture.

#### 5.2.2. Visual Comparison

Figure 6 shows reconstructions from all learned methods for an apple slice with bitter pit. The decrease in quality with the decrease in the number of angles is clearly visible. For 2 angles, none of the methods are able to accurately recover the shape of the apple. The iCTU-Net reconstruction has sharp edges for the 2-angle case, while the other methods produce blurry reconstructions.

The inner structure, including the defects, is accurately reconstructed for 50 angles by all methods. The only exception is the iCTU-Net. Reconstructions from this network show a smooth interior of the apple. The other methods also result in the disappearance of smaller defects with fewer measurement angles. Nonetheless, a defect-detection system might still be able to sort out the apple based on the 5-angle reconstructions. The 2-angle case can be used to assess failure modes of the different approaches. The undersampling case is so severe that a lot of information is lost. However, the iCTU-Net is able to produce a smooth image of an apple, but it has few similarities with the ground truth apple. It appears that the models have memorized the roundness of an apple and produce a round apple that has little in common with the real apple except for its size and core.

#### 5.2.3. Data Consistency

The data consistency is evaluated for all three Apple CT datasets. The MSE is used to measure the discrepancy. It is the canonical choice for measurements with Gaussian noise (cf. Section 4.1.3). Table A6 in the appendix contains all MSE values and standard deviations. Figure 7 shows the results depending on the number of angles for the noise-free and Gaussian noise dataset.

In the noise-free setup, the optimal MSE value is zero. Nonetheless, an optimal data consistency does not correspond to perfect reconstructions in this case. Due to the undersampling of the measurements, the discretized linear forward operator *A* has a non-trivial null space, that is, $\tilde{x}\in X$, apart from $\tilde{x}=0$, for which $A\tilde{x}=0$. Any element from the null space can be added to the true solution $x^{†}$ without changing the data discrepancy
$$\begin{array}{c} A\left(x^{†}+\tilde{x}\right)=Ax^{†}+A\tilde{x}=Ax^{†}+0=Ax^{†}=y. \end{array}$$

In the Gaussian setup, the MSE between noise-free and noisy measurements is used as a reference for a good data discrepancy. The problem from the undersampling is also relevant in this setting.

Both setups show an increase in the data discrepancy with fewer measurement angles. The reason for the increase is presumably the growing number of deviations in the reconstructions. In the Gaussian noise setup, the high data discrepancy of all learned methods for 2 angles coincides with the poor reconstructions of the apple slice in Figure 6. Only the TV method, which enforces data consistency during the reconstruction, keeps a constant level. The main problem for this approach are the ambiguous solutions due to the undersampling. The TV method is not able to identify the correct solution given by the ground truth. Therefore, the PSNR and SSIM values are also decreasing.

Likewise, the data consistency was analyzed for the dataset with scattering noise. The MSE values of all learned methods are close to the empirical noise level. In contrast, FBP and TV have a much smaller discrepancy. Therefore, their reconstructions are most likely influenced by the scattering noise. An effect that is also reflected in the PSNR and SSIM values in Table 4.

## 6. Discussion

Among all the methods we compared, there is no definite winner that is the best on both LoDoPaB-CT and Apple CT. Learned primal-dual, as an example of a learned iterative method, is the best method on LoDoPaB-CT, in terms of both PSNR and SSIM, and also gives promising results on Apple CT. However, it should be noted that the differences in performance between the learned methods are relatively small. The ISTA U-Net, second place in terms of PSNR on LoDoPaB-CT, scores only 0.14 dB less than learned primal-dual. The performance in terms of SSIM is even closer on LoDoPaB-CT. The best performing learned method resulted in an SSIM that was only 0.022 higher than the last placed learned method. The observation that the top scoring learned methods did not differ greatly in terms of performance has also been noted in the fastMRI challenge [61]. In addition to the performance of the learned methods, other characteristics are also of interest.

### 6.1. Computational Requirements and Reconstruction Speed

When discussing the computational requirements of deep learning methods, it is important to distinguish between training and inference. Training usually requires significantly more processing power and memory. All outputs of intermediate layers have to be stored for the determination of the gradients during backpropagation. Inference is much faster and less resource-intensive. In both cases, the requirements are directly influenced by image size, network architecture and batch size.

A key feature and advantage of the learned iterative methods, post-processing methods and fully-learned approaches is the speed of reconstruction. Once the network is trained, the reconstruction can be obtained by a simple forward pass of the model. Since the CINN, being a generative model, draws samples from the posterior distribution, many forward passes are necessary to well approximate the mean or other moments. Therefore, the quality of the reconstruction may depend on the number of forward passes [48]. The DIP + TV method requires a separate model to be trained to obtain a reconstruction. As a result, reconstruction is very time-consuming and resource-intensive, especially on the 972 px×972 px images in the Apple CT datasets. However, DIP + TV does not rely on a large, well-curated dataset of ground truth images and measurements. As an unsupervised method, only measurement data is necessary. The large size of the Apple CT images is also an issue for the other methods. In comparison to LoDoPaB-CT, the batch size had to be reduced significantly in order to train the learned models. This small batch size can cause instability in the training process, especially for CINN (cf. Figure A14).

#### Transfer to 3D Reconstruction

The reconstruction methods included in this study were evaluated based on the reconstruction of individual 2D slices. In real applications, however, the goal is often to obtain a 3D reconstruction of the volume. This can be realized with separate reconstructions of 2D slices, but (learned) methods might benefit from additional spatial information. On the other hand, a direct 3D reconstruction can have a high demand on the required computing power. This is especially valid when training neural networks.

One way to significantly reduce the memory consumption of backpropagation is to use invertible neural networks (INN). Due to the invertibility, the intermediate activations can be calculated directly and do not have to be stored in memory. INNs were successfully applied to 3D reconstructions tasks in MRI [62] and CT [63]. The CINN approach from our comparison can be adapted in a similar way for 3D data. In most post-processing methods, the U-Net can be replaced by an invertible iUnet, as proposed by Etmann et al. [63].

Another option is the simultaneous reconstruction of only a part of the volume. The information from multiple neighboring slices is used in this case, which is also referred to as 2.5D reconstruction. Networks that operate on this scenario usually have a mixture of 2D and 3D convolutional layers [64]. The goal is to strike a balance between the speed and memory advantage of the 2D scenario and the additional information from the third dimension. All deep learning methods included in this study would be suitable for 2.5D reconstruction with slight modifications to their network architecture.

Overall, 2.5D reconstruction can be seen as an intermediate step that can already be realized with many learned methods. The pure 3D case, on the other hand, requires specially adapted deep learning approaches. Technical innovations such as mixed floating point precision and increasing computing power may facilitate the transition in the coming years.

### 6.2. Impact of the Datasets

The type, composition and size of a dataset can have direct impact on the performance of the models. The observed effects can provide insight into how the models can be improved or how the results translate to other datasets.

#### 6.2.1. Number of Training Samples

A large dataset is often required to successfully train deep learning methods. In order to assess the impact of the number of data pairs on the performance of the methods, we consider the Apple CT datasets. The scattering noise dataset (Dataset C), with 5280 training images, is only about 10% as large as the noise free dataset (Dataset A) and the Gaussian noise dataset (Dataset B). Here it can be noted that the iCTU net, as an example of a fully learned approach, performs significantly worse on this smaller dataset than on dataset A and dataset B ( 26.26 dB PSNR on Dataset C with 50 angles, 36.07 dB and 32.90 dB on Dataset A and Dataset B with 50 angles, respectively). This drop in performance could also be caused by the noise case. However, Baguer et al. [34] have already noted in their work that the performance of fully learned approaches heavily depends on the number of training images. This could be explained by the fact that fully learned methods need to infer most of the information about the inversion process purely from data. Unlike learned iterative methods, such as learned primal-dual, fully learned approaches do not incorporate the physical model. A drop in performance due to a smaller training set was not observed for the other learned methods. However, 5280 training images is still comprehensive. Baguer et al. [34] also investigated the low-data regime on LoDoPaB-CT, down to around 30 training samples. In their experiments, learned primal-dual worked well in this scenario, but was surpassed by the DIP + TV approach. The U-Net post-processing lined up between learned Primal-Dual and the fully learned method. Therefore, the amount of available training data should be considered when choosing a model. To enlarge the training set, the DIP + TV approach can also be used to generate pseudo ground truth data. Afterwards, a supervised method with a fast reconstruction speed can be trained to mimic the behavior of DIP + TV.

#### 6.2.2. Observations on LoDoPaB-CT and Apple CT

The samples and CT setups differ greatly between the two datasets. The reconstructions obtained using the methods compared in this study reflect these differences to some extent, but there were also some effects that were observed for both datasets.

The sample reconstructions in Figure 4 and Figure 6 show that most learned methods produce smooth images. The same observation can be made for TV, where smoothness is an integral part of the modeling. An extension by a suitable regularization can help to preserve edges in the reconstruction without the loss of small details, or the introduction of additional noise. One possibility is to use diffusion filtering [65], for example, variants of the Perona-Malik diffusion [66] in this role. Diffusion filtering was also successfully applied as a post-processing step for CT [67]. Whether smoothness of reconstructions is desired depends on the application and further use of the images, for example, visual or computer-aided diagnosis, screening, treatment planning, or abnormality detection. For the apple scans, a subsequent task could be the detection of internal defects for sorting them into different grades. The quality of the reconstructions deteriorates with the decreasing number of measurement angles. Due to increasing undersampling, the methods have to interpolate more and more information to find an adequate solution. The model output is thereby influenced by the training dataset.

The effects of severe undersampling can be observed in the 2-angle setup in Figure 6. All reconstructions of the test sample show a prototypical apple with a round shape and a core in the center. The internal defects are not reproduced. One explanation is that supervised training aims to minimize the empirical risk on the ground truth images. Therefore, only memorizing and reconstructing common features in the dataset, like the roundness and the core, can be optimal in some ways to minimize the empirical risk on severely undersampled training data. Abnormalities in the data, such as internal defects, are not captured in this case. This effect is subsequently transferred to the reconstruction of test data. Hence, special attention should be paid to the composition of the training data. As shown in the next Section 6.2.3, this is particularly important when the specific features of interest are not well represented in the training set.

In the 5-angle setup, all methods are able to accurately reconstruct the shape of the apple. Internal defects are partially recovered only by the post-processing methods and the CINN. These approaches all use FBP reconstructions as a starting point. Therefore, they rely on the information that is extracted by the FBP. This can be useful in the case of defects but aggravating for artifacts in the FBP reconstruction. The CINN approach has the advantage of sampling from the space of possible solutions and the evaluability of the likelihood under the model. This information can help to decide whether objects in the reconstruction are really present.

In contrast, Learned Primal-Dual and the iCTU-Net work directly on the measurements. They are more flexible with respect to the extraction of information. However, this also means that the training objective strongly influences which aspects of the measurements are important for the model. Tweaking the objective or combining the training of a reconstruction and a detection model, that is, end-to-end learning or task-driven reconstruction, might be able to increase the model performance in certain applications [68,69].

#### 6.2.3. Robustness to Changes in the Scanning Setup

A known attribute of learned methods is that they can often only be applied to data similar to the training data. It is often unclear how a method trained in one setting generalizes to a different setting. In CT, such a situation could for example arise due to altered scan acquisition settings or application to other body regions. Switching between CT devices from different manufacturers can also have an impact.

As an example, we evaluated the U-Net on a different number of angles than it was trained on. The results of this experiment are shown in Table 5. In most setups the PSNR drops by at least 10 dB when evaluated on a different setting. In practice, the angular sampling pattern may change and it would be cumbersome to train a separate model for each pattern.

#### 6.2.4. Generalization to Other CT Setups

The LoDoPaB-CT and Apple CT datasets were acquired by simulating parallel-beam measurements, based on the Radon transform. This setup facilitates large-scale experiments with many example images, whereas the underlying operators in the algorithms have straightforward generalizations to other geometries. Real-world applications of CT are typically more complex. For example, the standard scanning geometries in medical applications are helical fan-beam or cone-beam [36]. In addition, the simulation model does not cover all physical effects that may occur during scanning. For this reason, the results can only be indicative of performance on real data.

However, learned methods are known to adapt well to other setups when retrained from scratch on new samples. It is often not necessary to adjust the architecture for this purpose, other than by replacing the forward operator and its adjoint where they are involved. For example, most learned methods show good performance on the scattering observations, whereas the classical methods perform worse compared to the Gaussian noise setup. This can be explained by the fact that the effect of scattering is structured, which, although adding to the instability of the reconstruction problem, can be learned to be (partially) compensated for. In contrast, classical methods require the reconstruction model to be manually adjusted in order to incorporate knowledge about the scattering. If scattering is treated like an unknown distortion (i.e., a kind of noise), such as in our comparison, the classical assumption of pixel-wise independence of the noise is violated by the non-local structure of the scattering. Convolutional neural networks are able to capture these non-local effects.

### 6.3. Conformance of Image Quality Scores and Requirements in Real Applications

The goal in tomographic imaging is to provide the expert with adequate information through a clearly interpretable reconstructed image. In a medical setting, this can be an accurate diagnosis or plan for an operation; and in an industrial setting, the image may be used for detection and identification of faults or defects as part of quality control.

PSNR and SSIM, among other image quality metrics, are commonly used in publications and data challenges [61] to evaluate the quality of reconstructed medical images [70]. However, there can be cases in which PSNR and SSIM are in a disagreement. Although not a huge difference, the results given in Table 4 are a good example of this. This often leads to the discussion of which metric is better suited for a certain application. The PSNR expresses a pixel-wise difference between the reconstructed image and its ground truth, whereas the SSIM checks for local structural similarities (cf. Section 4.1). A common issue with both metrics is that a local inaccuracy in the reconstructed image, such as a small artifact, would only have a minor influence on the final assessment. The effect of the artifact is further downplayed when the PSNR or SSIM values are averaged over the test samples. This is evident in some reconstructions from the DIP + TV approach, where an artifact was observed on multiple LoDoPaB-CT reconstructions whereas this is not reflected in the metrics. This artifact is highlighted with a red circle in the DIP + TV reconstruction in Figure A2.

An alternative or supporting metric to PSNR and SSIM is visual inspection of the reconstructions. A visual evaluation can be done, for example, through a blind study with assessments and rating of reconstructions by (medical) experts. However, due to the large amount of work involved, the scope of such an evaluation is often limited. The 2016 Low Dose CT Grand Challenge [9] based their comparison on the visibility of liver lesions, as evaluated by a group of physicians. Each physician had to rate 20 different cases. The fastMRI Challenge [61] employed radiologists to rank MRI reconstructions. The authors were able to draw parallels between the quantitative and blind study results, which revealed that, in their data challenge, SSIM was a reasonable estimate for the radiologists’ ranking of the images. In contrast, Mason et al. [71] found differences in their study between several image metrics and experts’ opinions on reconstructed MRI images.

In industrial settings, PSNR or related pixel-based image quality metrics fall short on assessing the accuracy or performance of a reconstruction method when physical and hardware-related factors in data acquisition play a role in the final reconstruction. These factors are not accurately reflected in the image quality metrics, and therefore the conclusions drawn may not always be applicable. An alternative practice is suggested in [72], in which reconstructions of a pack of glass beads are evaluated using pixel-based metrics, such as contrast-to-noise ratio (CNR), and pre-determined physical quantification techniques. The physical quantification is object-specific, and assessment is done by extracting a physical quality of the object and comparing this to a reference size or shape. In one of the case studies, the CNR values of iterated reconstructions suggest an earlier stopping for the best contrast in the image, whereas a visual inspection reveals the image with the “best contrast” to be too blurry and the bead un-segmentable. The Apple CT reconstructions can be assessed in a similar fashion, where we look at the overall shape of a healthy apple, as well as the shape and position of its pit.

### 6.4. Impact of Data Consistency

Checking the discrepancy between measurement and forward-projected reconstruction can provide additional insight into the quality of the reconstruction. Ground truth data is not needed in this case. However, an accurate model A of the measurement process must be known. Additionally, the evaluation must take into account the noise type and level, as well as the sampling ratio.

Out of all tested methods, only the TV, CGLS and DIP + TV approach use the discrepancy to the measurements as (part of) their minimization objective for the reconstruction process. Still, the experiments on LoDoPaB-CT and Apple CT showed data consistency on the test samples for most of the methods. Based on these observations, data consistency does not appear to be a problem with test samples coming from a comparable distribution to the training data. However, altering the scan setup can significantly reduce the reconstruction performance of learned methods (cf. Section 6.2.3). Verification of the data consistency can serve as an indicator without the need for ground truth data or continuous visual inspection.

Another problem can be the instability of some learned methods, which is also known under the generic term of adversarial attacks [73]. Recent works [74,75] show that some methods, for example, fully learned and post-processing approaches, can be unstable. Tiny perturbations in the measurements may result in severe artifacts in the reconstructions. Checking the data discrepancy may also help in this case. Nonetheless, severe artifacts were also found in some reconstructions from the DIP + TV method on LoDoPaB-CT.

All in all, including a data consistency objective in training (bi-directional loss), could further improve the results from learned approaches. Checking the discrepancy during the application of trained models can also provide additional confidence about the reconstructions’ accuracy.

### 6.5. Recommendations and Future Work

As many learned methods demonstrated similar performance in both low-dose CT and sparse-angle CT setups, further attributes have to be considered when selecting a learned method for a specific application. As discussed above, consideration should also be given to reconstruction speed, availability of training data, knowledge of the physical process, data consistency, and subsequent image analysis tasks. An overview can be found in Table 6. From the results of our comparison, some recommendations for the choice and further investigation of deep learning methods for CT reconstruction emerge.

Overall, the learned primal-dual approach proved to be a solid choice on the tested low photon count and sparse-angle datasets. The applicability of the method depends on the availability and fast evaluation of the forward and the adjoint operators. Both requirements were met for the 2D parallel beam simulation setup considered. However, without adjustments to the architecture, more complicated measurement procedures and especially 3D reconstruction could prove challenging. In contrast, the post-processing methods are more flexible, as they only rely on some (fast) initial reconstruction method. The performance of the included post-processing models was comparable to learned primal-dual. A disadvantage is the dependence on the information provided by the initial reconstruction.

The other methods included in this study are best suited for specific applications due to their characteristics. Fully learned methods do not require knowledge about the forward operator, but the necessary amount of training data is not available in many cases. The DIP + TV approach is on the other side of the spectrum, as it does not need any ground truth data. One downside is the slow reconstruction speed. However, faster reconstruction methods can be trained based on pseudo ground truth data created by DIP + TV. The CINN method allows for the evaluation of the likelihood of a reconstruction and can provide additional statistics from the sampling process. The invertible network architecture also enables the model to be trained in a memory-efficient way. The observed performance for 1000 samples per reconstruction was comparable to the post-processing methods. For time-critical applications, the number of samples would need to be lowered considerably, which can deteriorate the image quality.

In addition to the choice of model, the composition and amount of the training data also plays a significant role for supervised deep learning methods. The general difficulty of application to data that deviate from the training scenario was also observed in our comparison. Therefore, the training set should either contain examples of all expected cases or the model must be modified to include guarantees to work in divergent scenarios, such as different noise levels or number of angles. Special attention should also be directed to subsequent tasks. Adjusting the training objective or combining training with successive detection models can further increase the value of the reconstruction. Additionally, incorporating checks for the data consistency during training and/or reconstruction can help to detect and potentially prevent deviations in reconstruction quality. This potential is currently underutilized by many methods and could be a future improvement. Furthermore, the potential of additional regularization techniques to reduce the smoothness of reconstructions from learned methods should be investigated.

Our comparison lays the foundation for further research that is closer to real-world applications. Important points are the refinement of the simulation model, the use of real measurement data and the transition to fan-beam/cone-beam geometries. The move to 3D reconstruction techniques and the study of the influence of the additional spatial information is also an interesting aspect. Besides the refinement of the low photon count and sparse-angle setup, a future comparison should include limited-angle CT. A first application of this setting to Apple CT can be found in the dataset descriptor [38].

An important aspect of the comparison was the use of PSNR and SSIM image quality metrics to rate the produced reconstructions. In the future, this assessment should be supplemented by an additional evaluation of the reconstruction quality of some samples by (medical) professionals. A multi-stage blind study for the evaluation of unmarked reconstructions, including or excluding the (un)marked ground truth image, may provide additional insights.

Finally, a comparison is directly influenced by the selection of the included models. While we tested a broad range of different methods, there are still many missing types, for example, learned regularization [18] and null space networks [76]. We encourage readers to test additional reconstruction methods on the datasets from our comparison and submit reconstructions to the respective data challenge websites: (https://lodopab.grand-challenge.org/, last accessed: 1 March 2021) and (https://apples-ct.grand-challenge.org/, last accessed: 1 March 2021).

## 7. Conclusions

The goal of this work is to quantitatively compare learned, data-driven methods for image reconstruction. For this purpose, we organized two online data challenges, including a 10-day *kick-off event*, to give experts in this field the opportunity to benchmark their methods. In addition to this event, we evaluated some popular learned models independently. The appendix includes a thorough explanation and references to the methods used. We focused on two important applications of CT. With the LoDoPaB-CT dataset we simulated low-dose measurements and with the Apple CT datasets we included several sparse-angle setups. In order to ensure reproducibility, the source code of the methods, network parameters and the individual reconstruction are released. In comparison to the classical baseline (FBP and TV regularization) the data-driven methods are able to improve the quality of the CT reconstruction in both sparse-angle and low-dose settings. We observe that the top scoring methods, namely learned primal-dual and different post-processing approaches, perform similarly well in a variety of settings. Besides that, the applicability of deep learning-based models depends on the availability of training examples, prior knowledge about the physical system and requirements for the reconstruction speed.

## Figures and Tables

**Figure 1 jimaging-07-00044-f001:**
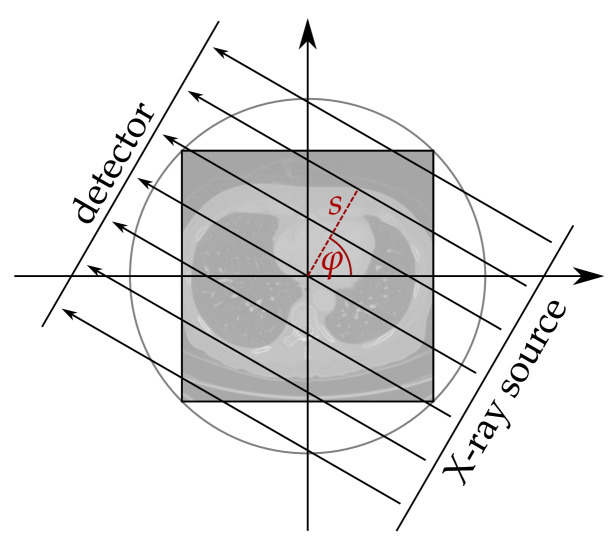
Parallel beam geometry. Adopted from [34].

**Figure 2 jimaging-07-00044-f002:**
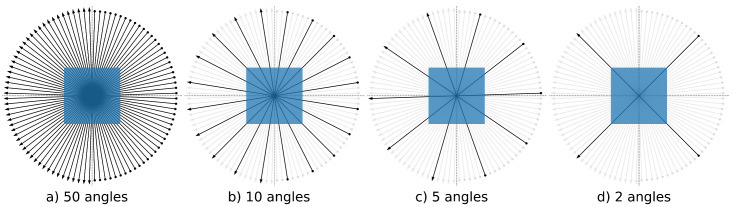
The angular sampling ranges employed for sparse image reconstructions for (**a**) 50 (full), (**b**) 10 (subset of 50 angles), (**c**) 5 (subset of 50 angles) and (**d**) 2 angles (subset of 10 angles). The black arrows show the position of the X-ray source (dot) and the position of the detector (arrowhead). For the sparse-angle scenario, the unused angles are shown in light gray.

**Figure 3 jimaging-07-00044-f003:**
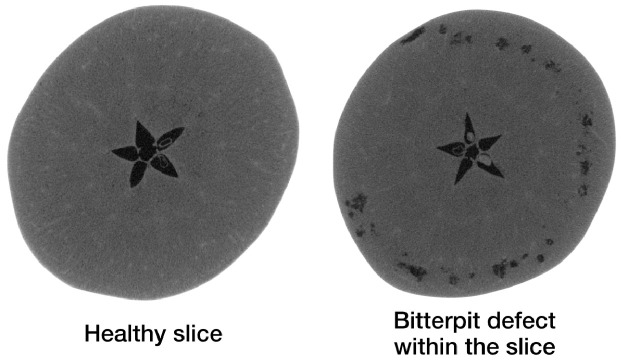
A horizontal cross-section of a healthy slice in an apple is shown on the **left**, and another cross-section with the bitter pit defects in the same apple on the **right**.

**Figure 4 jimaging-07-00044-f004:**
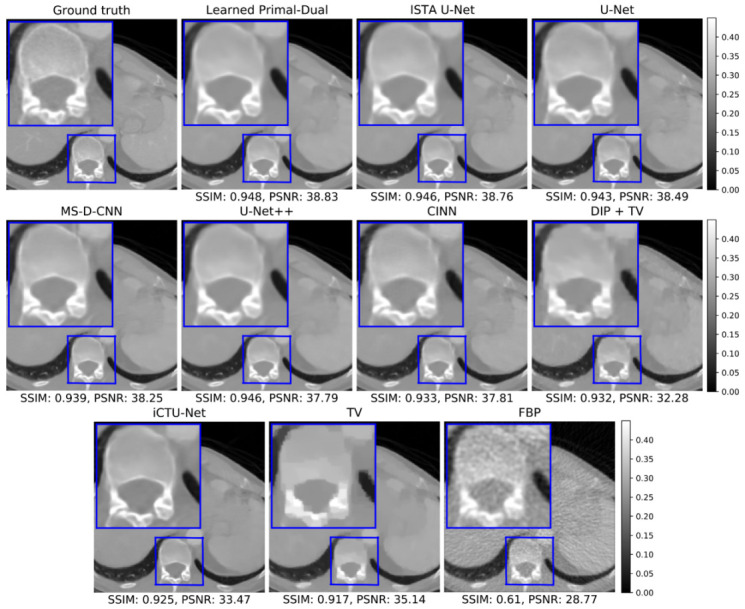
Reconstructions on the challenge set from the LoDoPaB-CT dataset. The window [0,0.45] corresponds to a HU range of ≈[−1001, 831] .

**Figure 5 jimaging-07-00044-f005:**
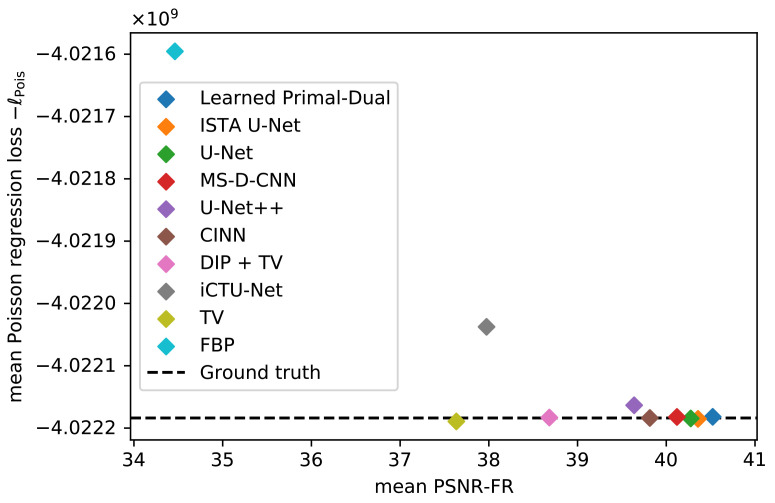
Mean data discrepancy $-\ell_{\mathrm{Pois}}$ between the noisy measurements and the forward-projected reconstructions, respectively the noise-free measurements. Evaluation is done on the LoDoPaB challenge images.

**Figure 6 jimaging-07-00044-f006:**
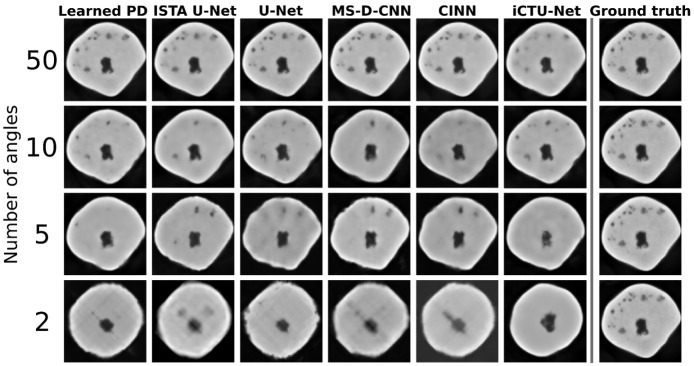
Visual overview of one apple slice with bitter pit for different learned methods. Evaluated on Gaussian noise. The quality of the reconstruction deteriorates very quickly for a reduced number of angles. For the 2-angle case, none of the methods can reconstruct the exact shape of the apple.

**Figure 7 jimaging-07-00044-f007:**
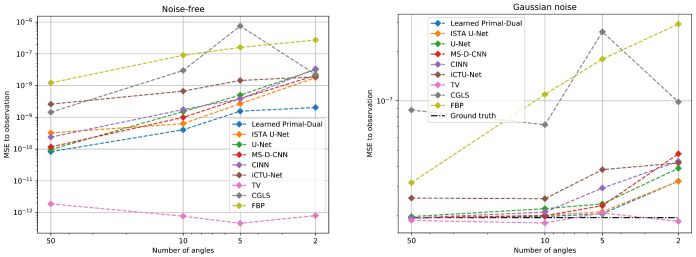
Mean squared error (MSE) data discrepancy between the measurements and the forward-projected reconstructions for the noise-free (**left**) and Gaussian noise (**right**) dataset. The MSE values are plotted against the number of angles used for the reconstruction. For the Gaussian dataset, the mean data discrepancy between noisy and noise-free measurements is given for reference. Evaluation is done on 100 Apple CT test images. See Table A6 for the exact values.

**Table 1 jimaging-07-00044-t001:** Settings of the low-dose parallel beam computed tomography (LoDoPaB-CT) and Apple CT datasets.

Property	LoDoPaB-CT	Apple CT
Subject	Human thorax	Apples
Scenario	low photon count	sparse-angle
Challenge	3678 reconstructions	100 reconstructions
Image size	362 px×362 px	972 px×972 px
Angles	1000	50, 10, 5, 2
Detector bins	513	1377
Sampling ratio	≈3.9	≈0.07–0.003

**Table 2 jimaging-07-00044-t002:** Results on the LoDoPaB-CT challenge set. Methods are ranked by their overall performance. The highest value for each metric is highlighted. All values are taken from the official challenge leaderboard https://lodopab.grand-challenge.org/evaluation/challenge/leaderboard/ (accessed on 4 January 2021).

Model	PSNR	PSNR-FR	SSIM	SSIM-FR	Number of Parameters
Learned P.-D.	36.25 ± 3.70	40.52 ± 3.64	0.866 ± 0.115	0.926 ± 0.076	874,980
ISTA U-Net	36.09 ± 3.69	40.36 ± 3.65	0.862 ± 0.120	0.924 ± 0.080	83,396,865
U-Net	36.00 ± 3.63	40.28 ± 3.59	0.862 ± 0.119	0.923 ± 0.079	613,322
MS-D-CNN	35.85 ± 3.60	40.12 ± 3.56	0.858 ± 0.122	0.921 ± 0.082	181,306
U-Net++	35.37 ± 3.36	39.64 ± 3.40	0.861 ± 0.119	0.923 ± 0.080	9,170,079
CINN	35.54 ± 3.51	39.81 ± 3.48	0.854 ± 0.122	0.919 ± 0.081	6,438,332
DIP + TV	34.41 ± 3.29	38.68 ± 3.29	0.845 ± 0.121	0.913 ± 0.082	hyperp.
iCTU-Net	33.70 ± 2.82	37.97 ± 2.79	0.844 ± 0.120	0.911 ± 0.081	147,116,792
TV	33.36 ± 2.74	37.63 ± 2.70	0.830 ± 0.121	0.903 ± 0.082	(hyperp.)
FBP	30.19 ± 2.55	34.46 ± 2.18	0.727 ± 0.127	0.836 ± 0.085	(hyperp.)

**Table 3 jimaging-07-00044-t003:** Mean and standard deviation of data discrepancy $-\ell_{\mathrm{Pois}}$. Evaluation is done on the LoDoPaB challenge images.

Method	$-\ell_{\mathrm{Pois}}\left(A\hat{x}\;|\;y_{\delta }\right)/10^{9}$
Learned Primal-Dual	−4.022182±0.699460
ISTA U-Net	−4.022185±0.699461
U-Net	−4.022185±0.699460
MS-D-CNN	−4.022182±0.699460
U-Net++	−4.022163±0.699461
CINN	−4.022184±0.699460
DIP + TV	−4.022183±0.699466
iCTU-Net	−4.022038±0.699430
TV	−4.022189±0.699463
FBP	−4.021595±0.699282
	$-\ell_{\mathrm{Pois}}\left(Ax^{†}\;|\;y_{\delta }\right)/10^{9}$
Ground truth	−4.022184±0.699461

**Table 4 jimaging-07-00044-t004:** Peak signal-to-noise ratio (PSNR) and structural similarity (SSIM) (adapted to the data range of each ground truth image) for the different noise settings on the Apple CT datasets. Best results are highlighted in gray. See Figure A7 and Figure A1 for a visualization.

Noise-Free	PSNR				SSIM			
Number of Angles	50	10	5	2	50	10	5	2
Learned Primal-Dual	38.72	35.85	30.79	22.00	0.901	0.870	0.827	0.740
ISTA U-Net	38.86	34.54	28.31	20.48	0.897	0.854	0.797	0.686
U-Net	39.62	33.51	27.77	19.78	0.913	0.803	0.803	0.676
MS-D-CNN	39.85	34.38	28.45	20.55	0.913	0.837	0.776	0.646
CINN	39.59	34.84	27.81	19.46	0.913	0.871	0.762	0.674
iCTU-Net	36.07	29.95	25.63	19.28	0.878	0.847	0.824	0.741
TV	39.27	29.00	22.04	15.95	0.915	0.783	0.607	0.661
CGLS	33.05	21.81	12.60	15.25	0.780	0.619	0.537	0.615
FBP	30.39	17.09	15.51	13.97	0.714	0.584	0.480	0.438
**Gaussian Noise**	**PSNR**				**SSIM**			
**Number of Angles**	**50**	**10**	**5**	**2**	**50**	**10**	**5**	**2**
Learned Primal-Dual	36.62	33.76	29.92	21.41	0.878	0.850	0.821	0.674
ISTA U-Net	36.04	33.55	28.48	20.71	0.871	0.851	0.811	0.690
U-Net	36.48	32.83	27.80	19.86	0.882	0.818	0.789	0.706
MS-D-CNN	36.67	33.20	27.98	19.88	0.883	0.831	0.748	0.633
CINN	36.77	31.88	26.57	19.99	0.888	0.771	0.722	0.637
iCTU-Net	32.90	29.76	24.67	19.44	0.848	0.837	0.801	0.747
TV	32.36	27.12	21.83	16.08	0.833	0.752	0.622	0.637
CGLS	27.36	21.09	14.90	15.11	0.767	0.624	0.553	0.616
FBP	27.88	17.09	15.51	13.97	0.695	0.583	0.480	0.438
**Scattering Noise**	**PSNR**				**SSIM**			
**Number of Angles**	**50**	**10**	**5**	**2**	**50**	**10**	**5**	**2**
Learned Primal-Dual	37.80	34.19	27.08	20.98	0.892	0.866	0.796	0.540
ISTA U-Net	35.94	32.33	27.41	19.95	0.881	0.820	0.763	0.676
U-Net	34.96	32.91	26.93	18.94	0.830	0.784	0.736	0.688
MS-D-CNN	38.04	33.51	27.73	20.19	0.899	0.818	0.757	0.635
CINN	38.56	34.08	28.04	19.14	0.915	0.863	0.839	0.754
iCTU-Net	26.26	22.85	21.25	18.32	0.838	0.796	0.792	0.765
TV	21.09	20.14	17.86	14.53	0.789	0.649	0.531	0.611
CGLS	20.84	18.28	14.02	14.18	0.789	0.618	0.547	0.625
FBP	21.01	15.80	14.26	13.06	0.754	0.573	0.475	0.433

**Table 5 jimaging-07-00044-t005:** Performance of a U-Net trained on the Apple CT dataset (scattering noise) and evaluated on different angular samplings. In general, a U-Net trained on a specific number of angles fails to produce good results on a different number of angles. PSNR and SSIM are calculated with image-dependent data range.

	Evaluation	50 Angles		10 Angles		5 Angles		2 Angles	
Training		PSNR	SSIM	PSNR	SSIM	PSNR	SSIM	PSNR	SSIM
	50 angles	39.62	0.913	16.39	0.457	11.93	0.359	8.760	0.252
	10 angles	27.59	0.689	33.51	0.803	18.44	0.607	9.220	0.394
	5 angles	24.51	0.708	26.19	0.736	27.77	0.803	11.85	0.549
	2 angles	15.57	0.487	14.59	0.440	15.94	0.514	19.78	0.676

**Table 6 jimaging-07-00044-t006:** Summary of selected reconstruction method features. The reconstruction error ratings reflect the average performance improvement in terms of the evaluated metrics PSNR and SSIM compared to filtered back-projection (FBP). Specifically, for LoDoPaB-CT improvement quotients are calculated for PSNR and SSIM, and the two are averaged; for the Apple CT experiments the quotients are determined by first averaging PSNR and SSIM values within each noise setting over the four angular sampling cases, next computing improvement quotients independently for all three noise settings and for PSNR and SSIM, and finally averaging over these six quotients. GPU memory values are compared for 1-sample batches.

Model	Reconstruction Error (Image Metrics)		Training Time	Recon- Struction Time	GPU Memory	Learned Para- Meters	Uses $D_{Y}$ Discre- Pancy	Operator Required
Learned P.-D.	🟉🟉	🟉	🟉🟉🟉🟉	🟉🟉	🟉🟉🟉🟉	🟉🟉	no	🟉🟉🟉
ISTA U-Net	🟉🟉	🟉	🟉🟉🟉	🟉🟉	🟉🟉🟉	🟉🟉🟉	no	🟉🟉
U-Net	🟉🟉	🟉	🟉🟉	🟉🟉	🟉🟉	🟉🟉	no	🟉🟉
MS-D-CNN	🟉🟉	🟉	🟉🟉🟉🟉	🟉🟉	🟉🟉	🟉	no	🟉🟉
U-Net++	🟉🟉	-	🟉🟉	🟉🟉	🟉🟉🟉	🟉🟉🟉	no	🟉🟉
CINN	🟉🟉	🟉	🟉🟉	🟉🟉🟉	🟉🟉🟉	🟉🟉🟉	no	🟉🟉
DIP + TV	🟉🟉🟉	-	-	🟉🟉🟉🟉	🟉🟉	3+	yes	🟉🟉🟉🟉
iCTU-Net	🟉🟉🟉	🟉🟉	🟉🟉	🟉🟉	🟉🟉🟉	🟉🟉🟉🟉	no	🟉
TV	🟉🟉🟉	🟉🟉🟉	-	🟉🟉🟉	🟉	3	yes	🟉🟉🟉🟉
CGLS	-	🟉🟉🟉🟉	-	🟉	🟉	1	yes	🟉🟉🟉🟉
FBP	🟉🟉🟉🟉	🟉🟉🟉🟉	-	🟉	🟉	2	no	🟉🟉🟉🟉
*Legend*	LoDoPaB	Apple CT	Rough values for Apple CT Dataset B					
	Avg. improv. over FBP		(varying for different setups and datasets)			
🟉🟉🟉🟉	0%	0–15%	>2 weeks	>10 min	>10 GiB	>$10^{8}$		Direct
🟉🟉🟉	12–16%	25–30%	>5 days	>30 s	>3 GiB	>$10^{6}$		In network
🟉🟉	17–20%	40–45%	>1 day	>0.1 s	>1.5 GiB	>$10^{5}$		For input
🟉		50–60%		≤0.02 s	≤1 GiB	≤$10^{5}$		Only concept

## Data Availability

The two datasets used in this study are the LoDoPaB-CT dataset [32], and the AppleCT dataset [37], both publicly available on Zenodo. The reconstructions discussed in Section 5 are provided as supplementary materials to this submission. These are shared via Zenodo through [57,58,59].

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
