# Peer review of "Quantitative Comparison of Deep Learning-Based Image Reconstruction Methods for Low-Dose and Sparse-Angle CT Applications"

_2313-433X, 2021, doi:10.3390/jimaging7030044_

Round 1
Reviewer 1 Report
The emergence of deep learning has led to several data-driven reconstruction methods for CT in recent years and a comprehensive comparison of these techniques on standardized datasets holds great value to theoreticians and practitioners alike. This paper presents a systematic evaluation of data-driven techniques for low-dose and sparse-view CT reconstruction on two public datasets. The paper also draws a comparison in terms of other important criteria such as the availability of training data, reconstruction time, etc.
I would consider this paper to be the first step towards standardizing the benchmarking process of different deep learning schemes for CT reconstruction. Nevertheless, the study is far from complete (which is kind of reflected in Table 6) and there are many unaddressed issues that need to be taken up as a part of future study.
Please find below some comments which I think should be taken into consideration to strengthen the technical contribution of the paper.
Section 1.1.1: Please clarify what you mean by the ‘ground-truth’. Does it mean reconstructed images from normal dose projections using classical methods? This is mentioned in Section 2.1, but it is not clear which reconstruction method was used for generating the ground-truth from normal-dose data. FBP or an iterative reconstruction scheme? The section also seems to have contradictory statements on the number of training examples in the LoDoPaB dataset.
Section 2.1: On what basis was the challenge set chosen? Was the training/testing division random?
After eq(2): Defining m as the product of s with phi is a bit of an abuse of notation, given how s and phi are defined in (1).
Section 3: The comparison does not include one important class of data-driven techniques, namely learned regularization schemes such as adversarial regularizers (reference [18]), network Tikhonov (Li et al., “NETT: solving inverse problems with deep neural networks”, Inverse Problems, vol. 36, 2020), etc. Is there any specific reason for that? These methods are particularly attractive as they enable the integration of deep learning within the classical regularization theory and should perhaps be included to make the study comprehensive.
Page 10: I fail to understand the rationale behind considering PSNR-FR as an evaluation metric. If L is substantially large than the intensity-range in the ground-truth, this will give a misleading measure of the quality of the reconstructed image. The same goes with SSIM-FR.
Section 4.1.3: Presumably, how one measures the data-discrepancy only seems to matter for TV and DIP, I suppose? The supervised methods are anyway trained to minimize a suitable distance measure (I am guessing squared-L2? Please mention it in Section 4.2.) between the network output and the target.
Table 2: I would suggest adding an extra column for the number of learnable parameters in each method. That will make it easier for a reader to compare the performance vis-à-vis model complexity. The numbers reported in Table 2 are also somewhat counter-intuitive in the sense that LPD (a method that incorporates the acquisition physics into the architecture) does not give any statistically significant improvement over post-processing schemes that are agnostic to the measurement physics. Could you explain why that is the case?
Section 5.1.2: “An exemplary reconstruction”---> “A representative reconstruction”
Section 5.1.2: “The learned methods produce somewhat smoother reconstructions in comparison to the ground truth.” This is probably because they approximate the posterior mean (which implicitly involves an averaging operation) when trained on squared-L2 loss. Perhaps it is worth mentioning somewhere. I believe the smoothness issue can be addressed by changing the training loss, and it perhaps merits a separate study.
Section 6.2.4: “However, learned methods are known to adapt well to other setups when retrained on new samples. It is often not necessary to adjust the architecture for this purpose.” Well, I am not quite convinced with this argument. LPD, for example, includes the forward operator in the architecture, and should probably be retrained from scratch when the acquisition geometry changes. In any case, it would be interesting to see an example of how LPD generalizes to a geometry different from what it was trained on. I would guess that a post-processing scheme would be more robust to such changes.
Author Response
Please see the attached PDF file.

Reviewer 2 Report
This paper presents interesting results on the quantitative comparison of different methods for image reconstruction presented during a data challenge.
In my opinion, the paper is suitable for publication.
I have only a few minor remarks:
Rows 9-10: PSNR and SSIM are not defined.
Rows 41-42: Authors declared: “Generally data-driven approaches promise fast and/or accurate image reconstruction by taking advantage of a large number of examples, i.e., training data.” More recent and fast deep learning networks were proposed in two different CT imaging studies, one for the segmentation of aneurysmal ascending aorta and one for the parenchyma segmentation in patients with idiopathic pulmonary fibrosis, namely ENET and ERFNET. They were initially used for image segmentation tasks in self-driving cars where hardware availability is limited and accurate segmentation is critical for user safety. They required far fewer images for training. In the opinion of the authors, could these methods represent a plausible solution for the image reconstruction task? Please integrate this consideration.
Section 1.1.1. The authors declared that the LoDoPaB-CT dataset contains more than 40000 images and successively they affirmed that it contains 35820 training images. I didn't understand the difference.
Author Response
Please see the attached PDF file.
